# Effectiveness of Plasma-Rich Fibrin and De-Epithelialized Free Gingival Graft in the Treatment of Gingival Recessions

**DOI:** 10.3390/medicina59030447

**Published:** 2023-02-23

**Authors:** Bojan Jovičić, Stevo Matijević, Stefan Veličković, Momir Stevanović, Aleksandra Mišić, Slavoljub Stanojević, Marija Bubalo

**Affiliations:** 1Dental Clinic, Military Medical Academy, University of Defence, 11040 Belgrade, Serbia; 2Department of Dentistry, Faculty of Medical Sciences, University of Kragujevac, 34000 Kragujevac, Serbia; 3Directorate of National Reference Laboratories, 11080 Belgrade, Serbia

**Keywords:** gingival recession, mucogingival surgery, soft tissue grafts, growth factors, plasma-rich fibrin, periodontal disease, periodontal attachment loss, surgical flap, tooth root, gingiva

## Abstract

*Introduction/Aim*: Soft tissue dehiscences such as gingival recessions are a very common problem that we face in modern periodontics. This clinical study aimed to analyze the effectiveness of surgical procedures using a de-epithelialized gingival graft (DGG) combined with a coronally advanced flap and to evaluate the application of plasma-rich fibrin (PRF). *Methods*: The study included 40 teeth (20 patients) with Miller class I and II gingival recessions. Twenty recessions (20 patients) were treated utilizing the de-epithelialized gingival graft in combination with the coronally advanced flap, and on the opposite side of the jaw, the same number of recessions were treated utilizing plasma-rich fibrin combined with the coronally advanced flap. To evaluate tissue condition and the clinical parameters before and after the surgical procedure, the following parameters were used: the degree of epithelial attachment (DEA), the width of keratinized gingiva (WKG), and the vertical depth of recession (VDR). *Results*: based on the achieved results and the analysis of clinical parameters, a statistically significant reduction in the vertical depth of recession was proven in both groups, with very similar mean percentages of root coverage, with the difference being that the stability of the soft tissues of the treated region was more visible in the DGG. *Conclusion*: using modern surgical procedures allows the regeneration of not only the soft tissues but also deeper periodontal tissues.

## 1. Introduction

Gingival recessions are defined as the apical migration of the marginal gingiva and belong to mucogingival deformities that represent anatomically functional irregularities resulting from an unfavorable correlation within the mucogingival complex consisting of epithelium, fur, cementum, and alveolar bone [1].

The particularity of gingival recessions is reflected in the multicausality of their etiological factors. Inadequate brushing techniques, high frenal attachment, an inadequate zone of the keratinized gingiva, a thin biotype of the soft tissue (less than 2 mm), as well as bone dehiscence are the main reasons for the occurrence of recessions [2]. However, recently, orthodontic treatment has been reported as the most common reason in almost 90% of cases with this anomaly [3].

Epidemiologically, the frequency of GR depends, to a large extent, on the age of the patient, but also certain favorable factors. Thus, the presence of gingival recessions (on one or more teeth) was determined in 58% of younger people and almost 100% of adults on at least one tooth. These values sufficiently indicate their importance.

The classification of gingival recessions is of great importance because it significantly helps us in making a prognosis before starting treatment. Today, there is a large number of different classifications, among which the most famous is the one according to Miller from 1986 [4]. It is divided into four classes depending on where the gingival edge is located concerning the mucogingival line, the presence of an interdental septum, and the occurrence of tooth malposition. The first and second classes are recessions that can be treated almost 100% of the time, while the third and especially the fourth classes represent major destruction and are associated with poor procedural outcomes [5]. One of the most widely used contemporary gingival recession classifications is the one proposed by Cairo in 2011 [6]. It is based on the condition of the underlying bone and CEJ loss at both the buccal and interproximal sites, and is divided into three categories depending on the condition of the CEJ. Type 1 shows no loss of interproximal attachment, and is clinically not detectable. Type 2 is associated with the loss of interproximal attachment. The amount of interproximal attachment lost is equal or less than the amount of buccal attachment lost. Type 3 recession is associated with the loss of interproximal attachment loss. The amount of interproximal attachment loss is greater than the attachment loss on the buccal side of the affected tooth.

Gingival recessions represent a problem for patients in the form of aesthetic deficiency, especially when it comes to the aesthetic zone, and the most common reason for patients to visit the dentist is the sensitivity to mechanical, thermal, and chemical stimuli of teeth where recessions are present [7].

Conservative treatment, in the sense of covering the exposed root with composite fillings, is considered a non-biological therapy and has no justification to be applied since there is an extremely large number of different surgical techniques that can successfully repair a large number of gingival recessions [8]. It is considered that almost 78% of untreated recessions, despite adequate oral hygiene, show a tendency to further progress [9].

The first surgical procedure for treating gingival recessions was published by Rosenthal back in 1912 [10]. At that time, it was considered a success to cover the exposed surface of the root as a purely quantitative aspect, while today, modern times and the patients themselves demand an absolute imitation of qualitative aspects, such as color, morphology, and tissue structure, that do not differ from the tissue of the neighboring region [11].

Various surgical techniques are used to cover the exposed surfaces of the tooth roots [12,13]. The purpose of surgical treatment is to fully restore function, to achieve the ideal aesthetics with maximum coverage of the exposed tooth root, to prevent further gingival retraction, and to obtain a wide and stable zone for the keratinized gingiva, which would also enable adequate plaque control and reduce tooth root hypersensitivity [14].

The application of the coronally displaced advanced flap (CAF) dates back to 30 years ago when it was first published as a technique, and today it is considered the gold standard when it comes to gum recession treatment [15]. The main reason for the application of the coronally advanced flap is to prevent the contraction and collapse of the flap after it is freed from the bone. Otherwise, CAF can be used alone or in conjunction with soft tissue grafts, collagen membranes, and growth factors (PRP; PRF; PRGF) [16].

Soft tissue or connective tissue grafts have been considered the golden standard for the past 50 years in periodontology and implantology due to their histological structure. They are used for two basic reasons, and those are the expansion of the zone of keratinized gingiva and the increase in the soft tissue volume [17]. A de-epithelialized gingival graft is an autograft of the connective tissue that is taken from the palate in the region of the first molar as a free gingival graft. After de-epithelialization, it is fixed in the desired position in the space of the recession that would have been previously prepared, and at the end of this procedure, it is completely covered with a coronally advanced flap of partial thickness [18].

In contrast to the application of soft tissue grafts, researchers in the last two decades have been devoted to a better understanding of processes at the cellular level and to a respect for tissue biology as a prerequisite for initiating more intensive regenerative processes [19]. The application of platelet-derived growth factors was implemented because platelets are the main cells that initiate the regeneration process and are the dominant cells when it comes to wound healing, which has been confirmed by numerous studies conducted in last decade of the last century [20]. Platelets in their alpha granules contain growth factors that are activated and released at the moment of platelet activation. Among these factors, PDGF (platelet-derived growth factors), TGF-Beta (transforming growth factor), VEGF (vascular endothelial growth factor), and IGF (insulin-like growth factor) stand out as the promoting factors of active tissue regeneration. Growth factors in periodontology and implantology are mostly used in the form of membranes [21]. Unlike platelet-rich plasma (PRP) or plasma rich in growth factors (PRGF), plasma-rich fibrin (PRF) is a completely natural blood derivative without anticoagulants and chemical additives [22]. The essence of the PRF method is precisely the fibrin and fibrin network that ensures the slow release of the growth factors over two weeks [23]. Moreover, fibrin, which is the essential component of growth factors, transforms into the host’s tissue over two weeks and stimulates regenerative processes through angiogenic stimulation. The main misconception, when growth factors were first introduced, was that they can create new bone and soft tissue. While there are only a few clinical trials that speak in favor of growth factors’ impact on hard tissue regeneration, many articles speak favorably of soft tissue reparation, specifically regarding soft tissue regeneration, stimulation, and maintenance of angiogenesis, which is the basis for rapid and successful tissue regeneration [24].

This work aimed to evaluate and compare the effect of two surgical procedures in solving the problem of gingival recession with the coverage of the exposed root surface and of the width of the zone of keratinized gingiva, a de-epithelialized gingival graft in combination with a coronally advanced flap of partial thickness and the PRF method in combination with a coronally advanced flap of full thickness, to determine the advantages and disadvantages of the applied techniques.

## 2. Materials and Methods

The examination within this comparative prospective study was performed on 20 patients, aged 20–45 years, who were non-smokers, who met the following criteria: the presence of gingival recession (class I and II according to Miller’s classification or class RT1 according to the EFP classification published in 2019) on canines with a visible and preserved cement–enamel junction on the buccal side of two teeth that were located in the same jaw contralaterally. The surfaces in the recession area had to be caries-free and intact on the buccal surface. All patients were treated at the Dental Clinic of the Military Medical Academy in Belgrade. On one side of the jaw, recessions were treated using a de-epithelialized gingival graft in combination with a coronally advanced flap of partial thickness (DGG group) (Figure 1). On the other side of the jaw, gingival recessions were treated with the application of growth factors—plasma-rich fibrin in combination with a coronally advanced flap of full thickness (PRF group) (Figure 2).

The assessment of the periodontal tissue state was evaluated by measuring the following parameters: the keratinized gingival width (KGW), the vertical dimension of recession (VDR), and the degree of epithelial attachment (DEA).

The state of the periodontal tissues was determined using clinical parameters, and so was the clinical state of the gingiva before the surgical procedure and six months after the surgery.

As mentioned above, soft tissue or connective tissue grafts have been considered the golden standard. Therefore, it is important to know and determine the reason for applying the grafts, whether the emphasis is on quantity, in terms of filling a soft tissue defect, or on quality, as is the case with the treatment of gingival recessions. The location in which grafts are harvested from the palate is also dependent on the aforementioned fact. Anatomically and histologically speaking, the region of the first premolar is the most voluminous and contains the fattiest tissue and glandular tissue and therefore provides an adequate volume if we want to focus on quantity [25]. Unlike this region, the part of the palate from the second premolar to the second molar contains more collagen and less fatty and glandular tissue. Therefore, this region is the primary site for harvesting grafts when treating gingival recessions. The de-epithelialized gingival graft is an autograft of the connective tissue that is taken from the palate in the region of the first molar as a free gingival graft, and after de-epithelialization, it is fixed in the desired position in the space of the recession that would have been previously prepared. At the end of this procedure, it is completely covered with a coronally advanced flap of partial thickness [26].

The method of graft harvesting was performed on the group in which the exposed root surface was treated with a de-epithelialized gingival graft (DGG), as described in detail by Zucchelli et al. [27]. It is performed under local anesthesia that is administered deeply into the vestibule, so as not to cause micro-injuries to the soft tissue in the region where the gingival recession is found. Before the first incision, the height of the recession is measured with a periodontal probe, and another 1 mm is added to the measured value to account for the expected contraction after surgery. After the measurement, the probe is transferred to the region of the papilla where the cut will be made and it is measured from the top of the papilla to the measured value where the first horizontal cut will be made. The same is done with the second papilla. Both incisions are made with a no. 15C blade. The length of the horizontal incision is on average within 2 to 3 mm, and it is important to ensure that there is a sufficiently wide base for the future papilla. After two horizontal partial-thickness incisions in the area of recession are made, two vertical shorter partial-thickness incisions are made up to the level of the apical edge of the gingival recession. The incision sites are both thoroughly irrigated with a 0.9% saline solution. With a fresh, unused no. 15C blade and at an angle of 70 degrees, the papilla is separated mesially and distally with a partial thickness up to the level of the apical edge of the recession. Doing this gave us medial and distal surgical papillae. From there, a full-thickness flap is separated above the root with the raspatory, but only until the bone above the recession is seen, which is approximately within 1–2 mm. After that, with a new no. 15 blade, a partial-thickness flap is separated so that the direction of the knife is parallel to the bone ridge. With this procedure, the mucosa is separated from the periosteum, and the incision goes deep until the mucosa transitions from immovable to mobile, i.e., to the mucogingival line. From there, the cut is still of partial thickness because the position of the knife is parallel to the soft tissue of the flap, which is why, in our study, we separated the muscles from the mucosa to allow the mobility of the entire lobe without making vertical incisions. After checking the tension and that the mobility of the flap is adequate, the graft is taken from the palate, again with an unused no. 15 blade. After local anesthesia is administered, the size of the graft is measured with a periodontal graduated probe, and the first horizontal incision is made approximately 1.5 mm away from the marginal edge of the tooth, followed by vertical incisions being made approximately 4 to 5 mm long in the depth of the palate. The direction of graft lifting goes from distal to mesial until the graft is completely separated from the base, after which two vertical incisions are joined with another horizontal one and the graft is completely removed from its base. After taking the graft, the blade is changed once again, and with another no. 15 blade, the epithelium is de-epithelialized from the surface of the graft to ensure adequate vascularization between the graft and the coronally advanced flap of partial thickness. De-epithelialization is performed by placing the harvested graft on a sterile and firm surface, and by placing the blade at a 45 degree angle in relation to the graft itself. The epithelium is removed in a slicing motion while the graft is irrigated with the 0.9% saline solution under adequate lighting. The epithelium should not be removed all at once, in order to preserve as much tissue as possible. A tamponade of the wound on the palate is performed with gauze soaked in a 0.9% saline solution. After that, the exposed root surface is treated with a curette. An amount of 24% EDTA is applied for 2 min to decontaminate the root surface and to open the dentine tubules, after which it is rinsed with saline solution. After the root treatment is completed, the papilla de-epithelialization is conducted with a no. 15C surgical knife in such a way as to leave a sufficient thickness of the connective tissue under the removed epithelium. By doing so, we obtained de-epithelialized anatomical papillae. The graft is placed over the exposed surface of the root and fixed in the desired position over the recession site at the base of the papilla. After that, the coronally advanced flap is also fixed over the graft. The graftmust be completely covered by the coronally advanced flap to allow for normal nutrition, and it is important to ensure the closest possible contact between the surgical and anatomical de-epithelialized papillae to achieve adequate revascularization of the treated region.

In the treatment of contralateral recession, the PRF method, described in detail by Choukroun et al., has been used in combination with the coronally advanced flap [28]. The surgical procedure for this is similar to that of the previous group, except that a full-thickness flap is raised here since PRF membranes are used as a type of matrix over the exposed root surface. Growth factors are obtained by taking a couple of tubes of venous blood, and usually, two to three tubes are enough, but sometimes more are required depending on the size of the defect. After extraction, they are placed in a centrifuge (the centrifuge we used was “Process PC 02 Designed by Joseph Choukroun”) at a certain number of revolutions, which in our case was 1400 turns in 14 min, which is the standard protocol for obtaining A-PRF. At the end of our centrifuge, three zones are separated in the test tube. The upper layer represents plasma that is poor in platelets, the middle layer is pure fibrin that is rich in growth factors, and the lower layer consists of erythrocytes. With tweezers, the central layer is taken and placed in a special box (PRF BOX) in which the fibrin clot is pressed and the membranes are obtained. This completes the preparation of the fibrin membrane, and its adaptation to the root surface occurs. The surgical procedure is completed by placing sutures and fixing the flap over the membranes in the coronal position.

Patients are given antibiotics within five to seven days and are advised to maintain hygiene, which includes only rinsing with 0.12% chlorhexidine for the first two weeks and brushing teeth with an ultra-soft brush for the next month.

Based on the established concept, six months after the operation, a postoperative assessment of the state of the periodontal tissues was carried out, and a definitive assessment of the initiated regenerative processes in the periodontium was carried out.

The obtained results were processed using appropriate statistical models: mean values, average changes in values, standard deviations, and Student’s *t*-test.

## 3. Results

The analysis of the results showed that the use of modern surgical techniques achieves a significant coverage of the exposed surfaces, an expansion of the zone of keratinized gingiva, as well as a reduction in the level of epithelial attachment, which creates the conditions for adequate oral hygiene. These techniques contribute to reducing all of the unpleasant sensations that the patients from the study had experienced before the surgical procedure. The values of VDR, as an important parameter in evaluating the success of gingival recession treatments, indicate that the application of growth factors (PRF) contributed to a significant reduction in gingival recession.

### 3.1. The Results of Implementing the PRF Technique

The preoperative value for VDR was 6.25 ± 0.48 mm, and six months later the value was 2.68 ± 0.25 mm (*t* = 5.58; *p* < 0.01). The difference of 3.57 mm in the VDR values shows the extent of coverage of the exposed roots.

The level of epithelial attachment is the most important parameter because it indicates the initiation of the regenerative processes. Thus, the average value before surgery (NPE) was 5.63 ± 0.32 mm, and after six months the value was 1.95 ± 0.19 mm (*t* = 4.36; *p* < 0.01).

The difference of 3.68 mm in the mean values for CAL represents the coronal dislocation of the epithelial attachment.

Differences are also evident in the zone of the keratinized gingiva (KG). The preoperative width of the keratinized gingiva was 1.38 ± 0.14 mm, and after six months it was 3.13 ± 0.42 mm (*t* = 5.36; *p* < 0.01). This means that an expansion of 1.75 mm of the zone of keratinized gingiva was achieved.

### 3.2. The Results Obtained by Implementing the Surgical Method

In the group of respondents where the de-epithelialized gingival graft (DGG) method was applied in combination with a coronally advanced flap of partial-thickness (DGG-group), statistically significant results were determined in terms of the examined parameters.

The preoperative average values for VDR, which was 6.17 ± 0.52 mm, and the value six months after the intervention, which was 0.37 ± 0.28 mm (*t* = 6.78; *p* < 0.01), indicate the importance of the applied treatment. The achieved coverage was about 95% of the exposed root surface.

Adequate vascularization, respecting the tissue biology of the treated region and the stimulation of the cells of the periodontal ligament caused the coronal dislocation of the epithelial attachment. The preoperative value was 5.55 ± 0.32 mm, and after six months it was 0.43 ± 0.36 (*t* = 6.56; *p* < 0.01), which is statistically very significant.

All of this caused the expansion of the zone of keratinized gingiva. The preoperative value was 1.24 ± 0.12 mm, and after 6 months it was 4.12 ± 0.46 mm (*t* = 5.18; *p* < 0.01), which shows that the zone of keratinized gingiva increased by 2.88 mm.

### 3.3. Comparative Analysis of Obtained Data

Table 1, Table 2 and Table 3 show the results of measuring the values of the indicators (parameters) of tissue condition (VDR, DEA, and WKG) before and after the intervention, as well as the results of the static analysis of the hypothesis testing. The attached results clearly show statistically significant differences in the obtained values of treatment efficiency indicators in both groups, using the PRF and DGG methods. The results of the mutual comparison of these two methods are also presented. The results of the comparison indicate that there are differences in the results obtained in the groups treated by different methods that are in favor of the DGG method (VDR: *t* = 12.8, *p* > 0.05; DEA: *t* = 11.8, *p* > 0.05; WKG: *t* = 3.3, *p* > 0.05).

The processing of the statistical data and the analysis of the average values of the results in both examined groups unequivocally show that the achieved results are due to the initiated regenerative processes in the periodontium, with certain advantages observed on the side where the graft was applied. The differences that can be observed are in the values for the zone of keratinized gingiva, the expansion of which ensures the stability of the soft tissue attachment, as well as the values for two other indicators of tissue condition, namely the vertical depth of the recession and epithelial attachment, which indicate the advantages of the procedure based on the de-epithelialized gingival graft.

## 4. Discussion

The essence of gingival recessions is not only the presence of an exposed root surface, which is visible clinically and which is an aesthetic problem for patients. Recessions are characterized by the loss of both soft tissue and cement, but also by the loss of the connective tissue attachment. This is precisely why it is important to initiate the regenerative processes, because otherwise, apart from the short-term covering of the roots, we will not achieve significant results, and the implications of this will be made visible very quickly by a relapse [29]. The purpose of the applied treatment is also to ensure the stability of the soft tissue attachment, which ensures the long-term maintenance of the achieved results, which is in concurrence with the research of Zucchelli and his co-authors, where they emphasize the significance of the quality of the epithelial attachment and the increase in the soft tissue volume [30].

The use of soft tissue grafts has long been accepted as the gold standard. However, the role of grafts in regenerative surgery has never been precisely defined before. Therefore, it is important to know and determine the reason for utilizing grafts, whether this is because of a focus on quantity, in terms of filling a soft tissue defect, or whether this is because of a focus on quality, as is the case of our approach to the treatment of gingival recessions. The location in which the graft is taken from the palate also depends on this [31,32,33]. Anatomically and histologically speaking, the region of the first premolar is the most voluminous, it mostly contains fatty and glandular tissue, and provides an adequate volume if we are focusing on quantity [34]. Unlike this region, the part of the palate from the second premolar to the second molar contains more collagen and less fat and glandular tissue, and therefore, it is the optimal graft harvesting site when it comes to treating gingival recessions [35].

A large number of authors, such as Zucchelli et al., but also Zuhr and Hurzeler, indicate the importance of the histological structure of the grafts, which largely depends on the harvesting site, as mentioned above [36].

The reason behind this is that grafts are living tissue that consist of living fibroblast cells, and the result of the applied surgery also depends on the technique of taking the graft, that is, the result depends on the part of the palate from which the graft is taken. Regarding the recessions and the covering of soft tissue defects, grafts harvested from the distal regions of the palate are favorable due to the fact that there is a prevailing presence of mature collagen tissue with dense collagen fibers and of a lamina propria without fatty or glandular tissue. To ensure rapid revascularization, and that is, for the graft to survive at the receiving site, it is necessary to establish the closest possible contact between the graft and the receiving site. Rejection of the graft occurs, among other things, in the case of the presence of fatty or glandular tissue. Carrying out the aforementioned step ensures that the regenerative processes are initiated. What is perhaps even more significant is that, as a function of time, by utilizing grafts, we achieve not only a better quality of soft tissue attachment, but also an increase in thickness, which is a change in the biotype of the gingiva, which decisively affects the stability of the achieved results [37]. Statistically significant thickening of the gingiva is attributed to the presence of the lamina propria, which is found in the connective tissue under the epithelium. In addition to all that, one study cited by Taveli et al. indicates that the use of DGG results in a better quality of keratinized tissue [38]. On the other hand, the use and importance of the growth factors should not be neglected, because it is only necessary to have a good knowledge of their action.

The results that have been achieved in our study unequivocally indicate that we have available methods that can successfully solve gingival recessions. Depending on the anatomical and morphological characteristics of the gingival recessions as well as the surgeon’s experience, the proper selection of surgical procedure is crucial in solving the most common mucogingival anomalies. When comparing the indicators of the gingival tissue condition of two groups of patients treated in parallel with the DGG and PRF techniques, we obtained statistically significant differences between these two groups. However, in both groups of patients, there was visible recovery of gingival tissue. Despite these statistical differences, in a clinical sense, taking into account the advantages of the procedure based on the PRF technique in terms of the comfort it provides to patients and the simplicity of its application, this method is equally good for routine use. Furthermore, the influence of these procedures on vascularization and their even stronger influence on the maintenance of angiogenesis are of immeasurable importance, which is pointed out by Choukroun and Simonpieri in their research [39]. Stimulating the formation of new blood vessels, especially in the first 24 h after surgery, is crucial for complete success. Therefore, the benefits of PRF application are irrefutable and they are the main reason for applyingthe growth factors in general [40]. It functions according to the principle that if there are preserved cells in a defect, there are chances for regeneration to start, otherwise we can only expect failure. The only difference between these two groups may be the function of time, because thanks to the histological structure of the grafts, we can expect a wider zone of keratinized gingiva and a thickening of the gingival biotype, which cannot be observed in the group where growth factors were applied, but the stability of the soft tissue attachment exists in cases where PRF is also utilized.

Also, any regenerative surgery, especially when it comes to soft tissue, is very unpredictable because we have to take a lot of factors into account, and perhaps the most important factor is vascularization. This is precisely the reason for the application of a partial-thickness flap and for the formation of a surgical or anatomical de-epithelialized papilla in the DGG group because the contact between these two papillae is central to the surgery, i.e., the key to vascularization and to the success of the surgery.

Regenerative treatment involves not only covering the root but also a step forward from the long epithelial attachment, as the most common outcome of the regenerative treatment, to the establishment of a functional connective tissue attachment, which is the central motive of utilizing connective tissue grafts as well as using growth factors as bio-mediators [41].

The importance of this research is that, by utilizing modern surgical procedures, we can start regenerative processes and ensure their stability over a longer period.

The success of the implemented therapy courses are greatly dependent on the proper maintenance of oral hygiene as well as a balanced dietary regimen. One beneficial aspect that can positively influence the outcomes of the procedures is the use of paraprobiotics and probiotics [42].

## 5. Conclusions

By applying modern methods, which include the use of connective tissue grafts, as well as a respect for tissue biology and the use of bio-mediators, it is possible to achieve both aesthetic and functional results in terms of the complete regeneration of the supporting tooth apparatus. In this way, we prevent the recurrence of recessions and ensure stable remission.

Based on the results before and after the surgical intervention, we can conclude that the results achieved using both methods indicate that the procedures were very successful and applicable. Despite the observed statistical differences that favor the DGG method over the PRF technique, taking into account the advantages of the latter in terms of the comfort it provides to patients and the simplicity of its application, the alternative PRF method is equally good for routine use. The main statistical difference between the two methods is in the amount of obtained keratinized gingiva, which can be attributed to the histological structure of the soft tissue grafts. A wider zone of keratinized gingiva ensures the long-term stability of achieved results.

## Figures and Tables

**Figure 1 medicina-59-00447-f001:**
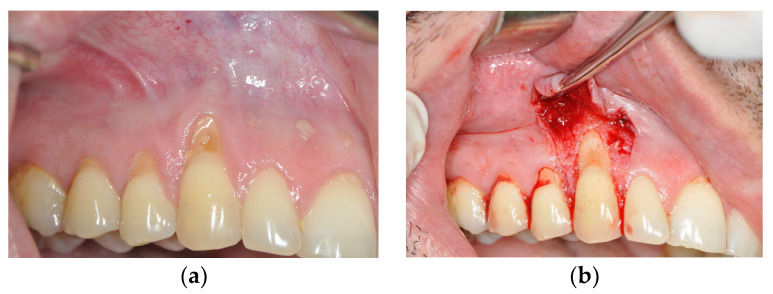
Gingival recession on tooth #13 treated with de-epithelialized gingival graft (DGG) and coronally advanced flap (CAF). (**a**) Gingival recession before the treatment; (**b**) trapezoidal incision made at the site of the recession; (**c**) DGG placed over the root at the site of the recession; (**d**) three months post-operation.

**Figure 2 medicina-59-00447-f002:**
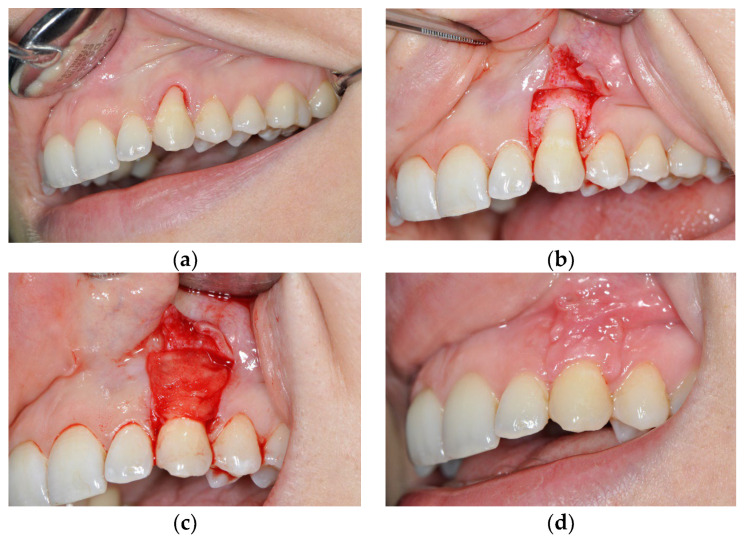
Gingival recession on tooth #23 treated with platelet rich fibrin (PRF) and coronally advanced flap (CAF). (**a**) Gingival recession before the treatment; (**b**) incision made at the site of the recession; (**c**) PRF placed at the site of the recession; (**d**) one month post-operation.

**Table 1 medicina-59-00447-t001:** Results of the statistical analysis of observed differences in tissue condition indicators (VDR) before and after the application of DGG and PRF procedures.

Statistics\Method Applied	PRF			DGG		
	Value	CI 95%		Value	CI 95%	
Average before the intervention, $\bar{x_{1}}$	6.25	5.77	6.73	6.17	5.65	6.69
Average after the intervention, $\bar{x_{2}}$	2.68	2.43	2.93	0.37	0.09	0.65
Mean difference, $\bar{x_{1}}$ − $\bar{x_{2}}$	3.57	3.03	4.11	5.80	5.21	6.39
Standard error of the mean, SEM_1_ (before intervention)	0.23			0.25		
Standard error of the mean, SEM_2_ (after intervention)	0.12			0.13		
Sample size	20			20		
Variance before the intervention, s_1_^2^	1.06			1.24		
Standard deviation before the intervention, s_1_	1.03			1.11		
Variance after the intervention, s_2_^2^	0.29			0.36		
Standard deviation after the intervention, s_2_	0.54			0.60		
Pooled variance, s_1,2_^2^_(pooled)_	0.67			0.80		
Difference within groups before and after intervention, *t*	5.6			6.8		
*p* value	<0.01			<0.01		
**Statistics\Method Applied**	**PRF vs. DGG (Comparative Differences between Two Applied Methods)**					
	**Value**				**CI 95%**	
Mean difference (before intervention), $\bar{x_{1}}$_(PRF)_ − $\bar{x_{1}}$_(DGG)_	0.08				−0.63	0.79
Mean difference (after intervention), $\bar{x_{2}}$_(PRF)_ − $\bar{x_{2}}$_(DGG)_	2.31				1.99	2.63
Pooled variance, s_1_^2^_PRF/DDG(pooled)_	1.15					
Difference between groups before intervention, *t*	0.2					
*p* value	<0.05					
Pooled variance, s_2_^2^ _(PRF/DDGpooled)_	0.32					
Difference between groups after intervention PRF vs. DGG, *t*	12.8					
*p* value	>0.05					

**Table 2 medicina-59-00447-t002:** Results of statistical analysis of observed differences in tissue condition indicators (DEA) before and after the application of DGG and PRF procedures.

Statistics\Method Applied	PRF			DGG		
	Value	CI 95%		Value	CI 95%	
Average before the intervention, $\bar{x_{1}}$	5.63	5.31	5.95	5.55	5.23	5.87
Average after the intervention, $\bar{x_{2}}$	1.95	1.76	2.14	0.43	0.24	0.62
Mean difference, $\bar{x_{1}}$ − $\bar{x_{2}}$	3.68	3.31	4.05	5.12	4.75	5.49
Standard error of the mean, SEM_1_ (before intervention)	0.15			0.15		
Standard error of the mean, SEM_2_ (after intervention)	0.09			0.09		
Sample size	20			20		
Variance before the intervention, s_1_^2^	0.47			0.47		
Standard deviation before the intervention, s_1_	0.69			0.69		
Variance after the intervention, s_2_^2^	0.17			0.17		
Standard deviation after the intervention, s_2_	0.41			0.41		
Pooled variance, s_1,2_^2^_(pooled)_	0.32			0.32		
Difference within groups before and after intervention, *t*	4.4			6.6		
*p* value	<0.01			<0.01		
**Statistics\Method Applied**	**PRF vs** **.** **DGG (Comparative Differences between Two Applied Methods)**					
	**Value**				**CI 95%**	
Mean difference (before intervention), $\bar{x_{1}}$_(PRF)_ − $\bar{x_{1}}$_(DGG)_	0.08				−0.37	0.53
Mean difference (after intervention), $\bar{x_{2}}$_(PRF)_ − $\bar{x_{2}}$_(DGG)_	1.52				1.12	1.19
Pooled variance, s_1_^2^_PRF/DDG(pooled)_	0.47					
Difference between groups before intervention, *t*	0.4					
*p* value	<0.05					
Pooled variance, s_2_^2^ _(PRF/DDGpooled)_	0.17					
Difference between groups after intervention PRF vs. DGG, *t*	11.8					
*p* value	>0.05					

**Table 3 medicina-59-00447-t003:** Results of statistical analysis of observed differences in tissue condition indicators (WKG) before and after the application of DGG and PRF procedures.

Statistics\Method Applied	PRF			DGG		
	Value	CI 95%		Value	CI 95%	
Average before the intervention, $\bar{x_{1}}$	1.38	1.24	1.52	1.24	1.12	1.36
Average after the intervention, $\bar{x_{2}}$	3.13	2.71	3.55	4.12	3.66	4.58
Mean difference, $\bar{x_{1}}$ − $\bar{x_{2}}$	1.75	1.31	2.19	2.88	2.40	3.36
Standard error of the mean, SEM_1_ (before intervention)	0.07			0.06		
Standard error of the mean, SEM_2_ (after intervention)	0.20			0.22		
Sample size	20			20		
Variance before the intervention, s_1_^2^	0.09			0.07		
Standard deviation before the intervention, s_1_	0.30			0.26		
Variance after the intervention, s_2_^2^	0.81			0.97		
Standard deviation after the intervention, s_2_	0.90			0.99		
Pooled variance, s_1,2_^2^_(pooled)_	0.45			0.52		
Difference within the group before and after intervention, *t*	5.4			5.2		
*p* value	<0.01			<0.01		
**Statistics\Method Applied**	**PRF vs. DGG (Comparative Differences between Two Applied Methods)**					
	**Value**				**CI 95%**	
Mean difference (before intervention), $\bar{x_{1}}$_(PRF)_ − $\bar{x_{1}}$_(DGG)_	0.14				−0.04	0.32
Mean difference (after intervention), $\bar{x_{2}}$_(PRF)_ − $\bar{x_{2}}$_(DGG)_	−0.99				−1.82	−0.16
Pooled variance, s_1_^2^_PRF/DDG(pooled)_	0.08					
Difference between groups before intervention, *t*	1.6					
*p* value	<0.05					
Pooled variance, s_2_^2^ _(PRF/DDGpooled)_	0.89					
Difference between groups after intervention PRF vs. DGG, *t*	−3.3					
*p* value	>0.05					

## Data Availability

Data can be provided upon request of the interested parties.

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
