# Peer review of "Effectiveness of Plasma-Rich Fibrin and De-Epithelialized Free Gingival Graft in the Treatment of Gingival Recessions"

_medicina, 2023, doi:10.3390/medicina59030447_

Round 1
Reviewer 1 Report
It was evaluated the original article A modern concept in gingival recessions treatment.
It is a clinical study and deserves respect (it is not simple developing clinical studies) . However, there is no novelty.
Below, I am sending my comments to improve the article.
———
TITLE: I considered it vague. Please, change it.
ABSTRACT
- lines 15-16, 19, 108, 133: please, it is completely wrong the word and terminology. The authors are using a technique; then, you need to understand what you are applying (“precisely plasma-rich fibrin (PRF))”
- results and conclusion need to be rewritten
INTRO: It can be reduced.
Please, I suggest to quote the following umbrella review recently published about the theme, in order to show a better knowledge and the state-of-the-art.
https://pubmed.ncbi.nlm.nih.gov/34898911/
- lines 36-39, 58-61, 67-70: include reference(s)
- lines 46-57: included refs. and use the current classification for gingival recessions
- line 81: “coronally displaced flap (CDF)”. It is understandable but it is not the correct nomenclature.
- lines 85-86: it is completely wrong. Correct it. “Otherwise, …, and growth factors (PRP; PRF; PRGF)”
- line 111: “Moreover, the growth factors are not the essence, but fibrin, which transforms”. Explain better this phrase
- Lines 114-115: “It is incorrect because the purpose of their application is to stimulate vascularization and maintain angiogenesis…”. Please, take care with this affirmation. It is partially correct. Improve the text to transmit correctly to the readers.
- line 119: pay attention! “deep-epithelialized”. Correct it.
M&M: The readability is poor. Improve it.
- where is the IRB approval?
- The authors used 1400 RPM and 14 min, which is a A-PRF technique. Please, clarify it in the text.
- What was the type of centrifuge (brand/company)?
- Line 126: “(class I and II according to Miller's classification)”. Update it for the current classification.
RESULTS: it is good. But I suggest to rewrite it improve the readability
DISCUSSION
Lines 319-321: I suggest to include the following articles in your discussion
https://pubmed.ncbi.nlm.nih.gov/35735651/
https://pubmed.ncbi.nlm.nih.gov/33527757/
https://pubmed.ncbi.nlm.nih.gov/34235906/
CONCLUSION
Be more specific, shorten it!
REFERENCES
Ref 25, 27, 30 need to be edited.
Author Response
Dear Sir or Madam,
I would like to thank you for your constructive criticism regarding my paper. I have taken your suggestions into consideration, and I have altered my paper accordingly. Please find my responses in the file attached, where I have adressed your suggestions point by point. I have uploaded the previous response by mistake instead of saving it as a draft. The final response will be uploaded here as a Word document. I would like to apologise once agan and thank you for understanding.
Kind regards,
Dr Bojan Jovicic
Point 1: TITLE: I considered it vague. Please, change it. –
Response 1. Thank you for pointing out our omission. Corrected. I have completely changed the title.
Point 2. ABSTRACT - lines 15-16, 19, 108, 133: please, it is completely wrong the word and terminology. The authors are using a technique; then, you need to understand what you are applying (“precisely plasma-rich fibrin (PRF))” - results and conclusion need to be rewritten.
Response 2. Thank you for pointing out our omission. We have adjusted the terminology and wording, and I have rewritten and shortened the results and conclusion to improve readability. Line 14-16 in Abstract the new text was added: “This clinical study aimed to analyzeevaluate the effectiveness of surgical procedures using de-epithelialized gingival graft (DGG) combined with a coronally displaced advanced flap and to evaluate the application of growth factors, more precisely plasma-rich fibrin (PRF)”, also in the rest of the manuscript the proper terminology was used.
Point 3. INTRO: It can be reduced.
Please, I suggest to quote the following umbrella review recently published about the theme, in order to show a better knowledge and the state-of-the-art.
https://pubmed.ncbi.nlm.nih.gov/34898911/
Response 3. Thank you for your comment, the new reference was added at number 22, in reference list and mentioned in the line 124
-Point 4. lines 36-39, 58-61, 67-70: include reference(s)
Response 4. Thank you for your comment. The new references were added at number 2, 7 and 10 in the reference list, and in the text at the lines 44-48, 70-73 and 81-83
- Point 5. lines 46-57: included refs. and use the current classification for gingival recessions
Response 5. Thank you for your comment. The new references and classification were added at line 57, 61 and 63, and in the reference list 4, 5 and 6
-Point 6. line 81: “coronally displaced flap (CDF)”. It is understandable but it is not the correct nomenclature.
Response 6: Thank you for your comment. The nomenclature was corrected throughout the whole manuscript.
- Point 7. lines 85-86: it is completely wrong. Correct it. “Otherwise, …, and growth factors (PRP; PRF; PRGF)”
Response 7: Thank you for your comment. The sentence was completely changed and is now at the line 99-101
- Point 8. line 111: “Moreover, the growth factors are not the essence, but fibrin, which transforms”. Explain better this phrase
Response 8: Thank you for your comment. The phrasing was corrected completely in order to be more clear. The corrected sentence is now at the line 126-129
- Point 9. Lines 114-115: “It is incorrect because the purpose of their application is to stimulate vascularization and maintain angiogenesis…”. Please, take care with this affirmation. It is partially correct. Improve the text to transmit correctly to the readers.
Response 9. Thank you for your comment. This sentence was also corrected in order not to sound polarizing. The improved text is now at the line 131-135
- Point 10. line 119: pay attention! “deep-epithelialized”. Correct it. - I have followed your instructions point by point and have included the suggested paper.
Response 10. Thank you for your comment. The nomenclature was corrected throughout the whole manuscript.
M&M: The readability is poor. Improve it.
- Point 11. where is the IRB approval?
Response 11. Thank you for your comment. The IRB approval and the ethics board approval was already provided and sent to the journal before submitting the manuscript.
- Point 12. The authors used 1400 RPM and 14 min, which is a A-PRF technique. Please, clarify it in the text.
Response 12. Thank you for your comment. The technique was more closely explained and is now in the line xx/xx
-Point 13. What was the type of centrifuge (brand/company)?
Response 13. Thank you for your comment. The type of centrifuge was also listed in the aforementioned sentence and is in the line 253-256
- Point 14. Line 126: “(class I and II according to Miller's classification)”. Update it for the current classification.
Response 14. Thank you for your comment. The classification was updated as mentioned above.
Point 15. RESULTS: it is good. But I suggest to rewrite it improve the readability
Response 15. Thank you for your comment. The results were rewritten and the methods and the comparative study were separated in order to improve the readability.
Point 16. DISCUSSION Lines 319-321: I suggest to include the following articles in your discussion
https://pubmed.ncbi.nlm.nih.gov/35735651/
https://pubmed.ncbi.nlm.nih.gov/33527757/
https://pubmed.ncbi.nlm.nih.gov/34235906/
Response 16. Thank you for your comment. The references that you provided were included, and are now at the reference list number 31, 32, 33 and in the line of text 358-359
Point 17. CONCLUSION Be more specific, shorten it!
Response 17. Thank you for your comment. The conclusion was rewritten and altered cosiderably in order to be more to the point.
Point 18. REFERENCES Ref 25, 27, 30 need to be edited.
Response 18. Thank you for your comment. The references were rewritten in order to update the citing style, and are now in the reference list under the numbers 34, 36, 39
Reviewer 2 Report
Manuscript of considerable interest for the dental sector, needs a major revision.
Abstract, to better highlight the results obtained.
Keywords, few and not present on MeSH, add more
Introduction: add the reference on the new classification of periodontal disease combined with the microbiological variation of a recessed gingiva
Materials and methods: poorly described, expand them
Very confusing results, reorganize them by highlighting the results obtained
Discussion: add as objectives the proactive action through the use of paraprobiotics and probiotics to maintain the result obtained for the clinical and microbiological management as already studied by the research group of Prof Scribante.
Conclusions: rephrase them based on the comments
Bibliography: add references required
Author Response
Dear Sir or Madam,
Thank you for your constructive criticism regarding my manuscript. I have carefully adressed all of your observations and have adjusted my manuscript accordingly. Please find the response in the attached document below.
Kind regards,
Dr Bojan Jovicic
Point nr 1. Abstract, to better highlight the results obtained.
Response 1. Thank you for your comment. The abstract was altered in order to improve the terminology.
Point nr. 2 Keywords, few and not present on MeSH, add more
Response 2. Thank you for your comment. These are the new added keywords: periodontal disease, PRF, periodontal attachment loss, surgical flap, tooth root, gingiva
Point nr. 3 Introduction: add the reference on the new classification of periodontal disease combined with the microbiological variation of a recessed gingiva
Response nr. 3 Thank you for your comment. The new classification was added at the reference nr. 4 and 6 and the text line 57 and 63
Point nr. 3 Materials and methods: poorly described, expand them
Response nr. 3 Thank you for your comment. The M&M were largely rewritten in order to better explained the used techniques and procedures.
Point nr 4. Very confusing results, reorganize them by highlighting the results obtained
Response nr. 4. The results were rewritten and the methods and the comparative study were separated in order to improve the readability.
Point nr. 5. Discussion: add as objectives the proactive action through the use of paraprobiotics and probiotics to maintain the result obtained for the clinical and microbiological management as already studied by the research group of Prof Scribante.
Response nr. 5. Thank you for your comment. The proactive use of paraprobiotics and probiotics was adressed, and the research paper was added as a reference at the line 423-426 and in the reference list under the nr. 42
Point nr. 6 Conclusions: rephrase them based on the comments
Response nr. 6. Thank you for your comment. The conclusion was almost completely rewritten in order to be shorter and to the point. It also adresses the points mentioned by the reviewers
Point nr. 7 Bibliography: add references required
Response nr. 7. Thank you for your comment. The references needed were added both in text and in the reference list. Please consult other responses for the complete alteration list.
Round 2
Reviewer 1 Report
Thank you for the revision.
I endorse the publication.
Reviewer 2 Report
The manuscript has been properly revised, it can be published.